# *Usp18* Expression in CD169^+^ Macrophages is Important for Strong Immune Response after Vaccination with VSV-EBOV

**DOI:** 10.3390/vaccines8010142

**Published:** 2020-03-23

**Authors:** Sarah-Kim Friedrich, Rosa Schmitz, Michael Bergerhausen, Judith Lang, Lamin B. Cham, Vikas Duhan, Dieter Häussinger, Cornelia Hardt, Marylyn Addo, Marco Prinz, Kenichi Asano, Philipp Alexander Lang, Karl Sebastian Lang

**Affiliations:** 1Institute of Immunology, University of Duisburg-Essen, Hufelandstrasse 55, 45147 Essen, Germany; Sarah-Kim.Friedrich@uk-essen.de (S.-K.F.); Rosa.Schmitz@gmx.de (R.S.); michael.bergerhausen@gmail.com (M.B.); Judith.Lang@uk-essen.de (J.L.); laminbcham@gmail.com (L.B.C.); Duhan.Vikas@uk-essen.de (V.D.); Cornelia.Hardt@uk-essen.de (C.H.); 2Clinic of Gastroenterology, Hepatology and Infectious Diseases, Heinrich-Heine-University, Moorenstrasse 5, 40225 Düsseldorf, Germany; haeussin@uni-duesseldorf.de; 3Division of Infectious Diseases, First Department of Medicine, University Medical Center Hamburg-Eppendorf, 20251 Hamburg, Germany; m.addo@uke.de; 4Department of Clinical Immunology of Infectious Diseases, Bernhard Nocht Institute for Tropical Medicine, 20359 Hamburg, Germany; 5German Center for Infection Research (DZIF), Partner Site Hamburg-Lübeck-Borstel-Riems, 20246 Hamburg Germany; 6Institute of Neuropathology, Medical Faculty, University of Freiburg, 79106 Freiburg, Germany; marco.prinz@uniklinik-freiburg.de; 7Signalling Research Centres BIOSS and CIBSS, University of Freiburg, 79106 Freiburg Germany; 8Center for Basics in NeuroModulation (NeuroModulBasics), Faculty of Medicine, University of Freiburg, 79106 Freiburg, Germany; 9Laboratory of Immune Regulation, School of Life Science, Tokyo University of Pharmacy and Life Sciences, Tokyo 192-0392, Japan; asanok@toyaku.ac.jp; 10Department of Molecular Medicine II, Medical Faculty, University Düsseldorf, Universitätsstrasse 1, 40225 Düsseldorf, Germany; langp@uni-duesseldorf.de

**Keywords:** enforced replication, vaccination, innate immunity, adaptive immunity, CD169^+^ macrophages, *Usp18*, Ebola virus, VSV-EBOV, Ervebo

## Abstract

Ebola virus epidemics can be effectively limited by the VSV-EBOV vaccine (Ervebo) due to its rapid protection abilities; however, side effects prevent the broad use of VSV-EBOV as vaccine. Mechanisms explaining the efficient immune activation after single injection with the VSV-EBOV vaccine remain mainly unknown. Here, using the clinically available VSV-EBOV vaccine (Ervebo), we show that the cell-intrinsic expression of the interferon-inhibitor *Usp18* in CD169^+^ macrophages is one important factor modulating the anti-Ebola virus immune response. The absence of *Usp18* in CD169^+^ macrophages led to the reduced local replication of VSV-EBOV followed by a diminished innate as well as adaptive immune response. In line, *CD169*-Cre^+/ki^ x *Usp18*^fl/fl^ mice showed reduced innate and adaptive immune responses against the VSV wildtype strain and died quickly after infection, suggesting that a lack of *Usp18* makes mice more susceptible to the side effects of the VSV vector. In conclusion, our study shows that *Usp18* expression in CD169^+^ macrophages is one important surrogate marker for effective vaccination against VSV-EBOV, and probably other VSV-based vaccines also.

## 1. Introduction

The Ebola virus (EBOV) is an enveloped, non-segmented negative RNA virus from the family of Filoviruses. It causes hemorrhagic fever in non-human primates and humans [1,2]. EBOV is one of the most pathogenic viruses worldwide, and epidemic outbreaks in the past, such as the Ebola virus outbreak in western Africa from 2013–2016, have demonstrated an urgent need for new vaccines in order to protect the human population. VSV-EBOV (rVSV-EBOV, Ervebo) is among the most powerful newly developed VSV vector vaccines; demonstrating safety and efficacy in clinical studies, and was recently approved by the EMA [3,4,5,6,7]. A single dose of immunization offers long-term protection in a B cell-dependent fashion [8]. Ervebo induces rapid protection after vaccination and shows its potential post-exposure efficacy [5]. However, the mechanisms which explain the effectiveness of the antiviral immune activation are basically unknown.

The splenic architecture builds a natural barrier against blood-borne pathogens [9]. A variety of factors are important to maintain these fragile structures and assure the functional synergy of spleen resident cells [10,11,12,13]. Due to their location in the marginal zone and proximity to blood conduits in the spleen, CD169^+^ macrophages (CD169^+^ MΦs) are the first line of defense against a variety of viruses [14,15,16,17]. During blood-borne infections, CD169^+^ MΦs are the first cell types to be infected. Therefore, CD169^+^ macrophages contribute to early antiviral immune activation [16,17]. Since CD169^+^ macrophages constitutively express high levels of the interferon-inhibitor *Usp18* (UBP43), they were postulated to accelerate viral replication [18]. Functionally, the ubiquitin-specific peptidase 18 (UBP43; *Usp18*) blocks IFNAR signaling through the competitive binding of JAK1 binding site at subunit IFNAR2, independent of its isopeptidase activity [19]. This results in a lack of Type-I interferon (IFN-I) production in CD169^+^ MΦs, and Honke et al. demonstrated that, therefore, VSV can undergo excessive replication in these cells. This mechanism is beneficial for the host, as it produces a sufficient amount of antigen for immune activation and subsequently leads to the strong production of IFN-I and is known as enforced virus replication [18]. IFN-Is are a group of inducible cytokines which can be produced by a variety of cell types, like leucocytes, endothelial cells and myeloid cells upon stimulation. The induction of antiviral pathways upon IFN-I binding leads to the induction of ISGs (interferon stimulated genes) like RNase L, RNA-dependent protein kinase (PKR) or Mx protein GTPases [20,21,22]. Fast and robust IFN-I production is not only essential for the induction of an antiviral state and clearance of viral infection, but also plays a major role in B cell activation and antibody response [23,24]. Neutralizing antibody responses are essential to control EBOV and VSV infections [25,26,27]. Whether the cell-intrinsic expression of *Usp18* in CD169^+^ macrophages indeed influences antiviral immune response remains unknown.

In this study, we demonstrate that enforced virus replication in CD169^+^ MΦs relies heavily on the intrinsic expression of *Usp18*. We show that enforced virus replication is essential for successful humoral responses during VSV-EBOV administration.

## 2. Material and Methods

### 2.1. Mice

*Usp18*^fl/fl^ mice were bred with mice expressing cre under the *CD169* promotor (*CD169*-cre) to generate *CD169*-Cre^+/*ki*^ x *Usp18*^fl/fl^ and *CD169*-Cre^+/+^ x *Usp18*^fl/fl^ mice. Usp18^fl/fl^ mice were generated by Marco Prinz [28]. *CD169*-cre mice were generated by Kenichi Asano [29]. *CD169*-Cre^+/*ki*^ x *Usp18*^fl/fl^ mice were compared with *CD169*-Cre^+/+^ x *Usp18*^fl/fl^ littermate controls. All animals were housed in single ventilated cages. Animal experiments were authorized by the Landesamt für Natur, Umwelt und Verbraucherschutz (LANUV) Nordrhein-Westfalen (Düsseldorf, Germany) and in accordance with the German law for animal protection and/or according to institutional guidelines at the Ontario Cancer Institute of the University Health Network. For survival experiments, animals were infected with a dose that was sublethal for WT animals. Animals were checked daily, killed with corresponding termination criteria, and counted as dead. Termination criteria were body weight, general condition, spontaneous behavior and clinical findings.

### 2.2. Virus

The VSV Indiana strain was originally obtained from D. Kolakofsky. VSV was propagated as described elsewhere [18]. VSV-EBOV (rVSV-EBOV, Ervebo) was obtained from M. Addo (Division of Infectious Diseases, First Department of Medicine, University Medical Center Hamburg-Eppendorf, Hamburg, Germany; Department of Clinical Immunology of Infectious Diseases, Bernhard Nocht Institute for Tropical Medicine, Hamburg, Germany; German Center for Infection Research, Partner Site Hamburg-Lübeck-Borstel-Riems). VSV-EBOV was propagated on BHK-21 cells. To analyze VSV-EBOV titers, 11 serial 1:3 dilutions of samples were prepared. Every second dilution was transferred to preseeded, confluent Vero E6 cells and overlaid after 2 h of incubation. After 48 h, plaque forming units were visualized by staining with crystal violet.

### 2.3. Neutralization Antibody Assay

VSV neutralization assay was performed as described elsewhere [18]. Briefly, neutralization assay samples were prediluted 1:40 (VSV) or 1:10 (VSV-EBOV). After complement inactivation at 56 °C for 30 min, serum was diluted 1:2 for 12 steps and incubated with 1 × 10^2^ PFU of VSV or VSV-EBOV for 90 min to obtain total Ig neutralization (VSV and VSV-EBOV). To analyze IgG neutralization, samples were preincubated with β-mercaptoethanol (0.1 M) to remove IgM before virus incubation (VSV). Samples were transferred to Vero E6 cells and overlaid with methylcellulose. of incubation, plaques were visualized via crystal violet staining.

### 2.4. Immunohistofluorescense

Organ samples were embedded in TissueTek O.T.C. (Sakura) medium and snap frozen in liquid nitrogen. 8 µm sections were generated. Sections were fixed with acetone and unspecific binding sites were blocked After 48 h using two percent fetal calf serum (FCS, Gibco) in PBS. VSV-Glycoprotein (VSV-G) was stained with Anti-VSV-G tag antibody (Abcam). Ebola virus glycoprotein was stained with Anti-Ebola surface glycoprotein (clone KZ52) from absolute antibodies. Secondary antibodies were obtained from ThermoFisher (anti-rabbit). Anti-CD169 (clone MOMA-1) was obtained from Bio-Rad. Antibodies were used in a dilution of 1:100 in blocking buffer and incubated for 30-60 minutes in a humidified darkened chamber at room temperature. Slides were covered with Fluorescence Mounting Medium (Dako, Glostrup, Denkmark) and processed for imaging with a Keyence BZ-9000 microscope.

### 2.5. RNA Extraction, cDNA Synthesis and qRT-PCR

RNA was extracted with Trizol Reagent (Invitrogen, Carlsbad, CA, USA) according to manufacturer’s protocol. Briefly, organs were homogenized in Trizol Reagent and phase separation was achieved by adding chloroform (Emsure, Merck, Darmstadt, Germany). RNA was precipitated by addition of isopropanol (Sigma, Saint Louis, MO, USA). After washing with 70% ethanol (Sigma, Saint Louis, MO, USA), the RNA pellet was dissolved in DEPC treated RNAse free water (Invitrogen, Carlsbad, California, USA). 400 ng of RNA were synthesized into cDNA with Quantitect Reverse Transcription kit (Qiagen, Hilden, Germany). Gene expression of *Ifna4* (QT01774353), *Ifna5* (QT00327656) and *Ifnb1* (QT00249662) was analyzed with primers obtained from Qiagen and normalized to Gapdh (QT01658692). VSV-NP expression was analyzed by primer sequence made in house.

### 2.6. Statistical Analysis

If not mentioned otherwise, data are expressed as arithmetic mean ± SEM and Student’s t-test was used to detect statistically significant differences (one- or two-tailed). P values of 0.05 or less were considered statistically significant. Statistical analyses and graphical presentations were computed with Graph Pad Prism software version 6 (Graph Pad, La Jolla, CA, USA).

## 3. Results

### 3.1. Usp18 Enforces Viral Replication in CD169^+^ Macrophages and Promotes Immune Activation

Previously we showed that the replication of cytopathic VSV in the spleen only occurs in the presence of UBP43 [18]. Mouse models with ubiquitous knockout of *Usp18* showed decreased virus replication and immune activation [18,30]. As *Usp18* also modulates Th17 cells and CD11b cells, it remained unknown whether a lack of *Usp18* in CD169^+^ cells indeed contributed to the anti-VSV immune response. To get insights, we used a mouse model with CD169^+^ cell-specific knockout of *Usp18* (*CD169*-Cre^+/*ki*^ x *Usp18*^fl/fl^*)*. We infected *CD169*-Cre^+/*ki*^ x *Usp18*^fl/fl^ systemically with VSV and analyzed virus replication after 16 h. VSV replication in inguinal draining lymph nodes (dLN) was below the detection limit in the absence of *Usp18* in CD169^+^ MΦs compared to WT conditions (Figure 1A). In line, analysis of VSV-NP expression in the spleen via qRT-PCR showed a significant reduction of the VSV-NP expression in *CD169*-Cre^+/*ki*^ x *Usp18*^fl/fl^ mice when compared to WT controls (Figure 1B). Visualization of the VSV antigen in an immunohistological analysis of spleen sections revealed strong colocalization of VSV-G (red) and CD169^+^ MΦs (blue) in WT mice in contrast to *CD169*-Cre^+/*ki*^ x *Usp18*^fl/fl^ mice, which did not show staining for VSV-G (Figure 1C). These data suggest that replication of VSV occurred and depended on *Usp18* in CD169^+^ macrophages.

We next aimed to determine whether *Usp18* is capable of activating immune responses upon systemic VSV infection [30]. In strong contrast to WT mice, *CD169*-Cre^+/*ki*^ x *Usp18*^fl/fl^ mice showed highly reduced IFNα serum levels following systemic VSV infection (Figure 1D). Consistently, the relative expression of *Ifna2*, *Ifna4* and *Ifnb1* was highly decreased in the spleen (Figure 1E) and LN (Figure 1F) 16 h post-infection in *CD169*-Cre^+/*ki*^ x *Usp18*^fl/fl^ mice. We concluded that enforced replication in CD169^+^ MΦs depended on *Usp18* and is necessary for a strong early immune response. Next, we addressed the involvement of *Usp18* in CD169^+^ macrophages during the activation of the adaptive immune response. The neutralizing antibody response in *CD169*-Cre^+/*ki*^ x *Usp18*^fl/fl^ was impaired when compared to control mice (Figure 1G,H). To further address the influence of *Usp18* expression in CD169^+^ MΦs in overall survival, we infected mice intravenously with a dose of VSV sublethal to WT mice. *CD169*-Cre^+/*ki*^ x *Usp18*^fl/fl^ mice were highly susceptible to VSV infection, which lead to the early death of these mice (Figure 1I).

Taken together, these data indicate that *Usp18* expression in CD169^+^ MΦs is crucial for enforced replication after systemic infection, leading to the strong activation of both innate and adaptive immune responses.

### 3.2. Enforced Replication Activates Immunity in dLN

We found that enforced VSV replication is important for the outcome of a systemic infection. In contrast, the administration of vaccines is often performed via intramuscular injection, directly targeting dLN [31]. To investigate whether the route of infection influenced the importance of *Usp18* in CD169^+^ macrophages, we infected WT and *CD169*-Cre^+/*ki*^ x *Usp18*^fl/fl^ mice subcutaneously with VSV and analyzed the viral replication in dLN after 16 h. Indeed, during subcutaneous infection, the presence of *Usp18* in CD169 MΦs was also essential for early replication (Figure 2A). In line with limited viral titers in *CD169*-Cre^+/*ki*^ x *Usp18*^fl/fl^ mice, the expression of VSV NP was dramatically decreased in *CD169*-Cre^+/*ki*^ x *Usp18*^fl/fl^ mice (Figure 2B).

Next, we analyzed the localization of VSV antigen in draining lymph nodes by histological analysis. WT mice showed a strong colocalization of VSV-G (red) and CD169^+^ MΦs (blue), whereas the expression of VSV-G was virtually absent in the *CD169*-Cre^+/*ki*^ x *Usp18*^fl/fl^ mice (Figure 2C). This indicates that early replication in draining lymph nodes is highly reliant on *Usp18* in CD169^+^ MΦs, as observed in systemic conditions.

Furthermore, we investigated whether enhanced replication can activate immune responses during local infection. We measured the relative expression of *Ifna4* and *Ifnb1* 16 h following subcutaneous VSV infection. The WT mice showed an increased expression of interferons compared to *CD169*-Cre^+/*ki*^ x *Usp18*^fl/fl^ mice (Figure 2D).

Next, we analyzed the role of *Usp18* during intramuscular immunization, as often applied during vaccination. As expected, the presence of *Usp18* improved the neutralizing Ab response in WT mice compared to *CD169*-Cre^+/*ki*^ x *Usp18*^fl/fl^ mice (Figure 2E,F). Together, these data show that during the local administration of VSV, *Usp18* in CD169^+^ macrophages is one important factor modulating the antiviral immune response.

### 3.3. Enforced Virus Replication Actives Immunity upon Systemic VSV-EBOV Administration

Next, we wondered whether vaccines based on a VSV vector similarly depend on *Usp18*. VSV-EBOV (Ervebo) is a powerful vaccine against the Ebola virus, which was engineered on the backbone of a WT VSV and the Glycoprotein of the Ebola Zaire virus. To get insights into the early immune events of VSV-EBOV administration, we infected mice systemically with VSV-EBOV. Seven hours after infection, WT mice showed enhanced viral titers in dLN compared to the *CD169*-Cre^+/*ki*^ x *Usp18*^fl/fl^ mice (Figure 3A). As viral titers in the spleen were below the detection limit, we performed a qRT-PCR analysis to quantify VSV-EBOV NP expression in the spleen and lymph nodes, which was strongly reduced in the *CD169*-Cre^+/*ki*^ x *Usp18*^fl/fl^ mice (Figure 3B). VSV-EBOV was detected in the spleens of WT mice, but hardly in the spleen of *CD169*-Cre^+/*ki*^ x *Usp18*^fl/fl^ mice (Figure 3C).

To investigate if this enhanced replication of VSV-EBOV also boosts immune responses, we analyzed innate immunity after 7 hours of infection. WT mice showed a stronger induction in serum IFNα levels (Figure 3D), as well as increased expression of *Ifna4* and *Ifnb1* in the spleen (Figure 3E) and lymph nodes (Figure 3F) compared to *CD169*-Cre^+/*ki*^ x *Usp18*^fl/fl */fl*^ mice. Furthermore, the induction of neutralizing antibodies was significantly higher in the WT mice (Figure 3G).

### 3.4. Enforced Virus Replication is Essential for Vaccination Success

Our findings show the importance of *Usp18* in immune activation after systemic challenge with VSV or VSV-EBOV. We next analyzed viral replication, localization and immune activation after subcutaneous injection of VSV-EBOV. In line with previous findings, *CD169*-Cre^+/*ki*^ x *Usp18*^fl/fl^ mice showed reduced VSV-EBOV titers (Figure 4A), as well as reduced VSV NP expression (Figure 4B) compared to WT mice. As expected, upon the immunohistological analysis of lymph node sections, staining for Ebola virus GP (red) colocalized with CD169^+^ MΦs (blue) (Figure 4C). In contrast, *CD169*-Cre^+/*ki*^ x *Usp18*^fl/fl^ did not show staining for Ebola virus GP. In line with these data, *CD169*-Cre^+/*ki*^ x *Usp18*^fl/fl^ showed a highly reduced induction of *Ifna4* and *Ifnb1* (Figure 4D).

To finally prove that enforced replication is essential for successful vaccination, we immunized mice intramuscularly with VSV-EBOV. The WT mice strongly induced a specificantibody response against VSV-EBOV, whereas *CD169*-Cre^+/*ki*^ x *Usp18*^fl/fl^ mice only showed low levels of VSV-EBOV-specific antibodies (Figure 4E).

To conclude, enforced virus replication mediated by *Usp18* in CD169^+^ MΦs is essential to induce an efficient vaccination status against VSV-EBOV in mice.

## 4. Discussion

The immune response after the injection of a live virus is a highly balanced system. Over-stimulated immune responses may participate in the development of autoimmunity, neurodegenerative diseases or other inflammatory processes [32,33,34,35,36]. On the other hand, a minimal immune induction might result in limited immune protection, and/or increased susceptibility of the host [37]. Therefore, the amplification of virus particles and accompanied immune responses have to be counterbalanced. Especially for the *Rhabdoviridae* representative VSV, the early events of immune response play an important role [27].

CD169^+^ MΦs, as the first cell types being infected during blood-borne infections, are crucial gatekeepers [14,15,16,17]. In the present study, we highlighted the distinct role of *Usp18* in CD169^+^ macrophages during natural infection and its potential role during vaccination. UBP43 is a multifunctional molecule involved in several processes. Beside its protease activity which controls cell metabolism, the turnover of transport proteins or the activation of transcription, *Usp18* is essential for cell development [38]. Absence of *Usp18* impaired the generation of CD11b^+^ dendritic cells, as well as Th17 cells [39]. Former mouse models investigated enforced virus replication in the ubiquitous absence of *Usp18* [30]. Due to its previously mentioned features, studies could not clarify if enforced replication solely relies on *Usp18* expression in CD169^+^ macrophages or is influenced by other cell types. For the first time, we were able to generate a model with CD169-specific knockout of *Usp18*. We found that the early replication of VSV is restricted to CD169^+^ macrophages and is strongly enhanced in the presence of *Usp18*. A strong activation of innate as well as adaptive immune responses is the result, which is essential to circumvent lethal infection.

VSV-EBOV (Ervebo) displays a recently developed VSV-vectored vaccine expressing the Ebola virus glycoprotein. Little is known about its behavior in systemic infection. Former models of VSV vectors with expression of filovirus glycoproteins, stated phenotypical identity of vectors and WT rhabdoviruses [40]. From previous experiments with WT VSV, we hypothesized that VSV-EBOV needs to replicate to induce its antiviral protection. Indeed, we found here that WT VSV and VSV-EBOV behave similarly. A strong *Usp18*-dependent replication restricted to CD169^+^ MΦs was observed in both cases, leading to a powerful induction of innate and adaptive immune responses. Interestingly, new insights underlined that CD169 may also serve as an entry factor for EBOV in humans, which strengthens the finding from our data that CD169-expressing cells contribute to immune responses after VSV-EBOV challenge [41].

CD169^+^ macrophages do not only play a unique role during blood-borne infections. Iannacone et al. showed that lymph node-resident CD169^+^ MΦs are crucial for preventing VSV from CNS invasion during subcutaneous infection in a type I interferon-dependent fashion [16]. Studies with live attenuated Rabies virus stated that intramuscular injection results in the transient infection of draining lymph nodes [31]. Based on these findings, we demonstrated that enforced VSV replication in dLN is arbitrative to induce IFN type I, B cell response and prevent the global spread of VSV.

Recent studies show that the early protection of VSV-EBOV is dependent on a strong activation of the innate immune response—mainly type I interferon induction. In line, we found strong induction of type I interferon in WT mice. Absence of *Usp18* in CD169^+^ macrophages blunted this strong IFN-I response. Therefore, we would suggest that this could be one factor making patients susceptible to side effects. This hints toward a dual advantage of enforced replication, as it accelerates vaccination success, but also limits the induction of side effects. Successful live vaccination does not only rely on early immune events; moreover, a strong memory guarantees long-term immunity against infections [42]. A robust B cell memory is considered especially indispensable, since live vaccination non-responders are associated with low neutralizing Ab titers [43]. According to this, a recent study in macaques stresses that the protective memory of the VSV-EBOV vaccine relies heavily on B cell response rather than cytotoxic T cell response [44]. Our findings underline a robust B cell response against VSV-EBOV as the result of enforced replication in dLNs which induces a successful vaccination status in mice.

Besides the VSV-EBOV (Ervebo) vaccine, viral vaccine vectors on the backbone of the VSV have risen over the last decade [45,46]. Here, we show that VSV-based vector vaccines are ideal vehicles for immunization with special regard to enforced replication in CD169^+^ macrophages. Type I interferon is essential to prevent the lethality of VSV infections in immunocompetent mice [47,48]. Therefore, VSV replication is only enhanced in cells with deprived IFN signaling, like CD169^+^ MΦs. It is possible that the enforced replication of VSV-based vector vaccines would not only boost immunity, but could also lessen immunization-associated adverse effects by restricting replication to CD169^+^ macrophages.

## 5. Conclusions

In summary, we have demonstrated that enforced virus replication in CD169^+^ MΦs is essential for activating immunity after VSV and VSV-EBOV infection, and is highly *Usp18*-dependent. Furthermore, we showed that the vaccination success of live vaccines is highly reliant on enforced replication.

## Figures and Tables

**Figure 1 vaccines-08-00142-f001:**
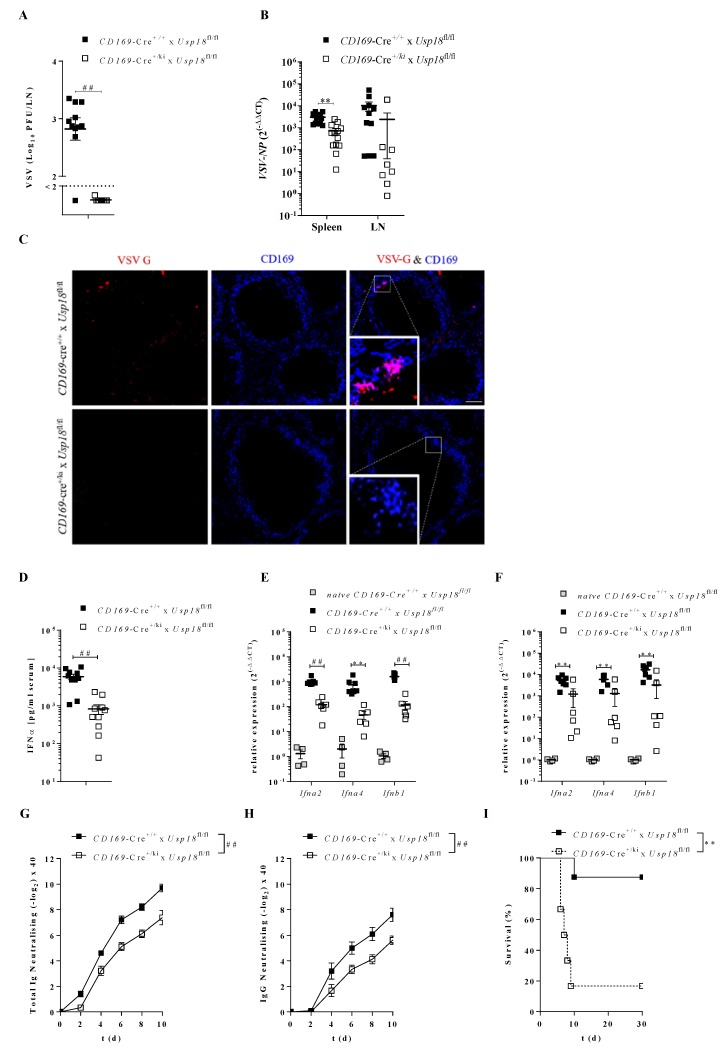
Enforced virus replication of VSV depends on *USP18* expression in CD169^+^ macrophages. **A**: Viral titers in lymph node 16 h after 2 × 10^6^ PFU i.v. VSV infection. (*n* = 9–11). **B**: Quantitative detection of VSV-NP in spleen (*n* = 13–14) and lymph node (*n* = 8–11) after infection as in **A**. **C**: Immunohistofluorescense staining of spleen sections of mice infected with 2 × 10^6^ PFU VSV i.v. 16 h p.i. stained for VSV-G (red) and CD169 (blue). Scale bar 100 µm. One representative picture of each group from two experiments (*n* = 7–8) shown. **D**: IFNα serum levels of mice infected with 2 × 10^6^ PFU VSV i.v. 16 h p.i. (*n* = 10). **E** + **F**: Relative expression of *Ifna2, Ifna4* and *Ifnb1* in spleen (**E**, *n* = 6–7) or lymph node (**F**, 6–7) obtained from infected mice as described in **A**. (*naïve* WT mice *n* = 4). **G** + **H**: VSV neutralizing total Ig (**G**) or neutralizing IgG (**H**) in mice i.v. infected with 2 × 10^6^ PFU VSV. (*n* = 9–10). **I**: Survival of mice i.v. infected with 2 × 10^6^ PFU. (*n* = 6–8) ** *p* < 0.01 and ^##^
*p* < 0.0001 (Student’s t-test, **A**,**B**,**D**–**F**), (Two-way-Anova, **G**,**H**), (Mantel-Cox test, **I**). Data are representative of two (**A–I**) experiments (mean ± SEM (**A**,**B**,**D**–**H**)).

**Figure 2 vaccines-08-00142-f002:**
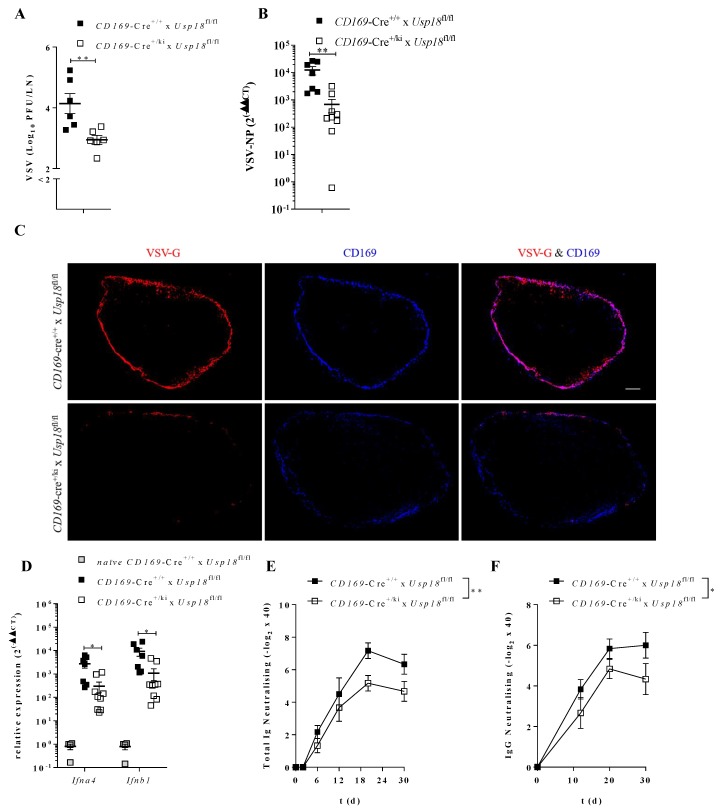
Enforced virus replication in dLN is essential to activate immunity. **A**: Viral titers in dLN 16 h after 2 × 10^6^ PFU s.c. VSV infection. (*n* = 6). **B**: Quantitative detection of VSV-NP in dLN after infection as in **A**. (*n* = 7–9). **C**: Immunohistofluorescense staining of dLN sections of mice infected with 2 × 10^6^ PFU VSV s.c. 16 h p.i. stained for VSV-G (red) and CD169 (blue). Scale bar 100 µm. One representative picture of each group from two experiments (*n* = 7–9) shown. **D**: Relative expression of *Ifna2, Ifna4* and *Ifnb1* in draining lymph node obtained from infected mice as described in **A**. (*n* = 7–9, *naïve* WT mice *n* = 4). **E + F**: VSV neutralizing total IG (**E**) or neutralizing IgG (**F**) in mice subcutaneously infected with 2 × 10^6^ PFU VSV. (*n* = 6–9) * *p* < 0.05 and ** *p* < 0.01 (Student’s t-test, **A**,**B**,**D**), (Two-way-Anova, **E**,**F**). Data are representative of two (**A–F**) experiments (mean ± SEM (**A**,**B**,**D**–**F**)).

**Figure 3 vaccines-08-00142-f003:**
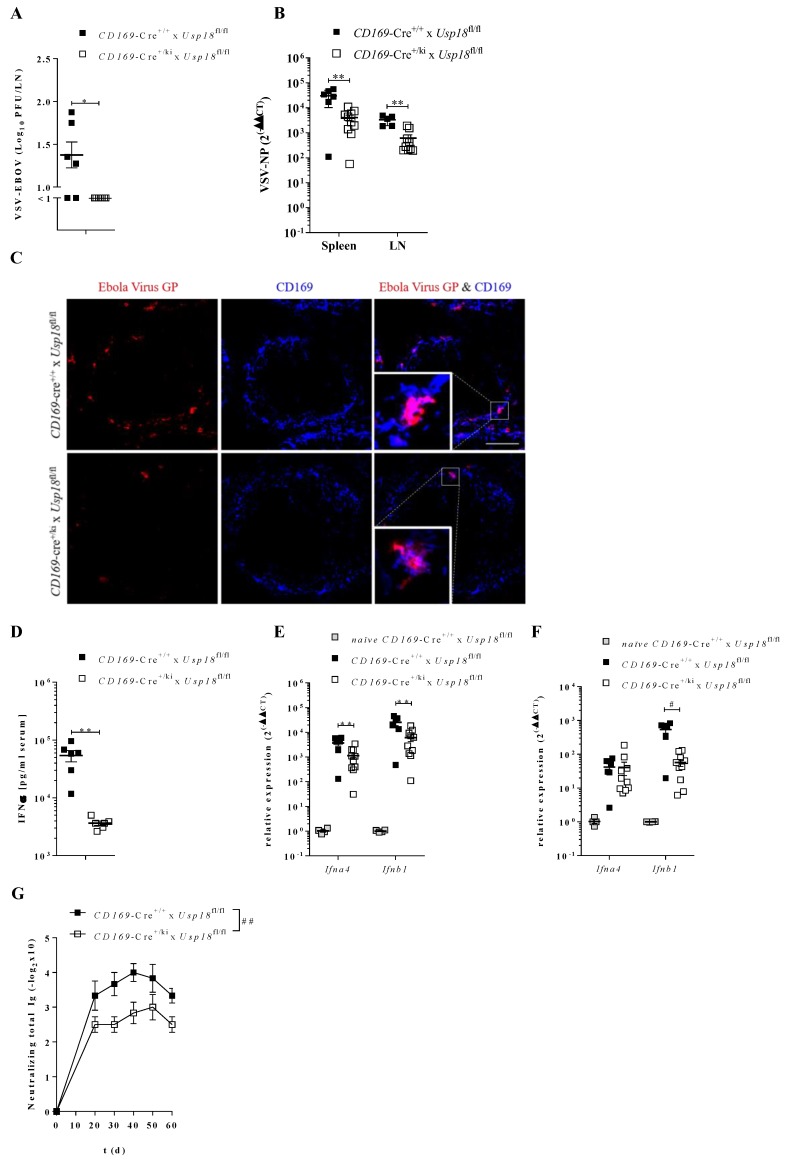
Enforced replication activates innate and adaptive immune response upon VSV-EBOV infection. **A**: Virus titers in LN 7 hours after 3 × 10^6^ PFU i.v. VSV-EBOV infection. (*n* = 6). **B**: Quantitative detection of VSV-NP in spleen and lymph node after infection as in **A**. (*n* = 6–10). **C**: Immunohistofluorescense staining of spleen sections of mice infected with 3 × 10^6^ PFU VSV-EBOV i.v. 7 hours p.i. stained for Ebola virus GP (red) and CD169 (blue). Scale bar 100 µm. One representative picture show from each group of two independent experiments (*n* = 6). **D**: IFNα serum levels of mice infected with 3 × 10^6^ PFU VSV-EBOV i.v. 16 h p.i. (*n* = 6). **E**,**F**: Relative expression of *Ifna4* or *Ifnb1* spleen (**E**, *n* = 6–10)) or lymph node (**F**, *n* = 6–10) obtained from infected mice as described in **A**. (*naïve* WT mice *n* = 4). **G:** VSV-EBOV neutralizing antibodies in serum of mice i.v. infected with 3 × 10^6^ PFU VSV-EBOV. (*n* = 6) * *p* < 0.05, ** *p* < 0.01, ^#^
*p* < 0.001 and ^##^
*p* < 0.0001 (Student’s t-test, **A**,**B**,**D**,**F**), (Two-way-Anova, **G**). Data are representative of two (**A**–**G**) experiments (mean ± SEM (**A**,**B**,**D**–**G**)).

**Figure 4 vaccines-08-00142-f004:**
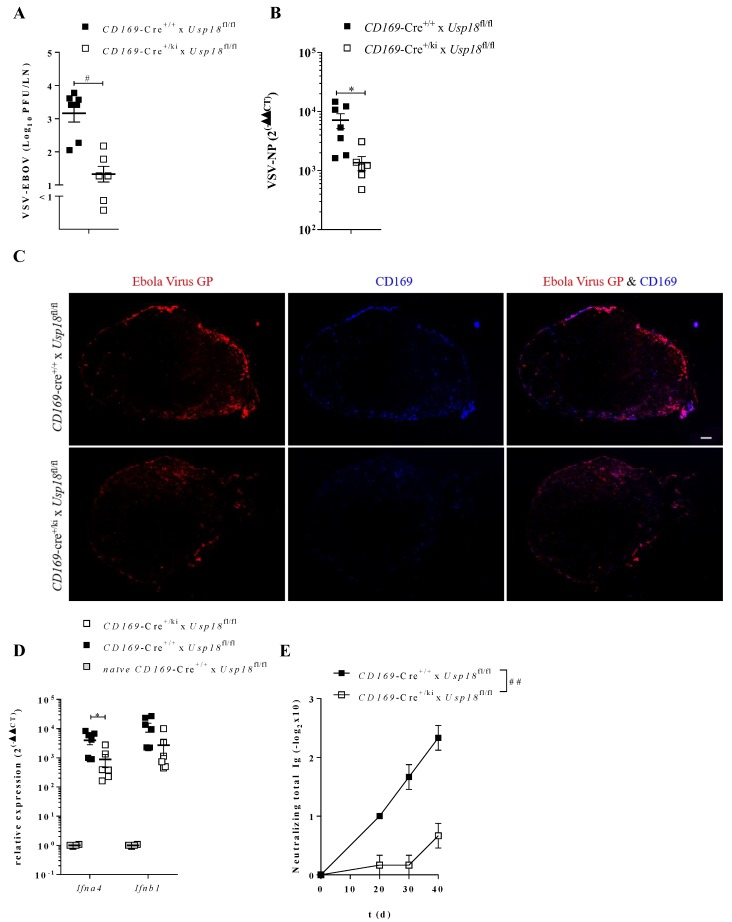
Enforced replication is essential for vaccination success of VSV-EBOV administration. **A** + **B**: VSV-EBOV titers (**A**, *n* = 6–7) or relative VSV-NP expression (**B**, *n* = 6–7) of WT and *CD169*-Cre^+/*ki*^ x *Usp18*^fl/fl^ mice in draining lymph nodes of mice subcutaneously infected with 2 × 10^6^ PFU VSV-EBOV 16 h p.i. **C**: Immunofluorescence of lymph node sections from WT and *CD169*-Cre^+/*ki*^ x *Usp18*^fl/fl^ mice 16 h after subcutaneous injection of 2 × 10^6^ PFU VSV-EBOV. Staining for Ebola virus GP (red) and CD169 (blue). One representative staining per group of two independent experiments (*n* = 6–7). **D**: Relative expression of *Ifna4* and *Ifnb1* in draining lymph nodes after infection as described in **A**. (*n* = 6–7, *naïve* WT mice *n* = 4). **E**: VSV-EBOV neutralizing total Ig in mice i.m. infected with 2 × 10^6^ PFU VSV-EBOV. (*n* = 6) ** p* < 0.05, ^#^
*p* < 0.001, ^##^
*p* < 0.0001 (Student’s *t*-test, **A**,**B**,**D**), (Two-way-Anova, **E**). Data are representative of two (**A**–**E**) experiments (mean ± SEM (**A,B,D,E**)).

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
