# Peer review of "Usp18 Expression in CD169+ Macrophages is Important for Strong Immune Response after Vaccination with VSV-EBOV"

_vaccines, 2020, doi:10.3390/vaccines8010142_

Round 1

Reviewer 1 Report

In this manuscript the authors use mice lacking USP18 expression in CD169+ macrophages to examine immune responses elicited by Vesicular Stomatitis virus (VSV) and by VSV-EBOV, the recently approved VSV-vectored Ebola virus vaccine. The manuscript describes a fairly straightforward story, using their mouse model to study various measures of response following infection with the different viruses and different immunization routes. There are, however, quite a number of grammatical changes that should be made in the text – I’ve tried to list some of these below, but there are more that should be addressed – once addressed, I think this should manuscript should be published.

Minor points:

I was unfamiliar with the term “enforced replication”, it may be worth defining what is meant by this in the introduction. Also in the introduction, please mention that the VSV-EBOV vaccine, Ervebo, is now approved for use.

Line 75: Is the VSV-EBOV used here exactly that used for Ervebo? If not, how does it differ? The authors indicate that VSV-EBOV was obtained from M. Addo; I’m not familiar with this person, please indicate their affiliation after their name. Are they with Merck, the manufacturer of Ervebo?

Line 232: “life virus” should be “live virus”

Line 252: “VSV-EBOV displays a recently developed vector vaccine on the backbone of VSV.” I think this should be something like “VSV-EBOV is a recently developed VSV-vectored vaccine displaying the EBOV glycoprotein.”

Line 255: “..VSV-EBOV needs to replicate for inducing its antiviral protection.” Should be “VSV-EBOV needs to replicate to induce its antiviral protection.”

Line 259: I don’t think that this paper (reference 32) demonstrates that “…CD169 serves as an entry receptor for EBOV in human…”, there are several pieces of data missing for that to be concluded. For example, there is no work with actual EBOV described, it’s all with virus-like particles. I suggest changing this sentence to “…CD169 may serve as an entry factor for EBOV in humans.”

Line 284: Sentence starting with “It is possible,..”, suggest changing to “It is possible that enforced replication of VSV-based vaccines not only boosts immunity but could also reduce immunization-associated adverse effects by restricting replication to CD169+ macrophages.”

Reviewer 2 Report

The manuscript by Friedrich et al studies the role of Usp18 expression in CD169+ macrophages during infection with vesicular stomatitis virus (VSV) or the recently FDA-approved VSV-EBOV vaccine. By corss-breeding CD169-Cre mice with Usp18fl/fl mice, the authors specifically knocked out Usp18 in CD169+ macrophages and evaluated their viral loads, and the induction of innate and adaptive immune responses following systemic and localized injections of VSV or VSV-EBOV. The authors have built upon their previous work demonstrating key role of Usp18 in enforced virus replication. The study is pertinent, well-done and provides a key mechanistic understanding of immune response to the Ebola vaccine. Authors have presented their data nicely and their conclusions are well supported by the data.

Major points:

  1. Authors have a very short introduction. They should expand it to explain the idea of “enforced virus replication”, cite relevant references for the basic EBOV information, introduce Usp18, and role of type I IFNs, role of antibody responses in protection against VSV and EBOV as all these topics will set the relevance of the results section for the reader.
  2. Materials and Methods are also very short and very limited in details. Especially the sections 2.3, 2.4 and 2.5 need more details. For example, there is no explanation of how “Total Ig neutralizing” and “IgG neutralizing titers” were estimated. The authors have generated beautiful immunofluorescence images, but it is almost impossible to gather enough protocol information from their methods section. Similarly, more details are also needed for section 2.5. Multiple times, authors have tested draining lymph nodes. Please specify which ones.
  3. Figure legends for Fig 2 are actually duplicated from Fig 1. Please fix.
  4. Figs 1c and 3c - Please specify which part of the image is zoomed in in the inset images.
  5. Line 111 - “NP expression via” should be “NP expression in spleen via”.
  6. The conventional abbreviation for VSV glycoprotein is G, not GP. Please fix it throughout the manuscript.

Minor points:

  1. Please check for grammatical errors and typos. Some examples are (not an exhaustive list): 
    • Line 104 - “Usp18 did not only show” should that be “Usp18 did not show”?
    • Line 152 - “VSV particles” should be “VSV antigen”
    • Line 187 - “To rule out” Is that what authors intended to say?
    • Line 232 - “life virus” should be “live virus”
    • Line 251 - “inevitable for” should be “inevitable to”
